# Association of Impaired Fasting Glucose and Diabetes with SARS-CoV-2 Spike Antibody Titers after the BNT162b2 Vaccine among Health Care Workers in a Tertiary Hospital in Japan

**DOI:** 10.3390/vaccines10050776

**Published:** 2022-05-13

**Authors:** Zobida Islam, Shohei Yamamoto, Tetsuya Mizoue, Akihito Tanaka, Yusuke Oshiro, Natsumi Inamura, Maki Konishi, Mitsuru Ozeki, Wataru Sugiura, Norio Ohmagari

**Affiliations:** 1Department of Epidemiology and Prevention, Center for Clinical Sciences, National Center for Global Health and Medicine, Tokyo 162-8655, Japan; syamamoto@hosp.ncgm.go.jp (S.Y.); mizoue@hosp.ncgm.go.jp (T.M.); mkonishi@hosp.ncgm.go.jp (M.K.); 2Department of Laboratory Testing, Center Hospital of the National Center for Global Health and Medicine, Tokyo 162-8655, Japan; tanaka.akihito.aj@mail.hosp.go.jp (A.T.); yuoshiro@hosp.ncgm.go.jp (Y.O.); ninamura@hosp.ncgm.go.jp (N.I.); miozeki@hosp.ncgm.go.jp (M.O.); 3Center for Clinical Sciences, National Center for Global Health and Medicine, Tokyo 162-8655, Japan; wsugiura@hosp.ncgm.go.jp; 4Center Hospital of the National Center for Global Health and Medicine, Tokyo 162-8655, Japan; nohmagari@hosp.ncgm.go.jp

**Keywords:** diabetes, impaired fasting glucose, COVID-19, SARS-CoV-2, vaccine, immunogenicity

## Abstract

Background: Hyperglycemia can alter the activation of innate and acquired immunity, but epidemiological evidence linking hyperglycemia to post-vaccination immunogenicity is limited. Objective: To examine the association between SARS-CoV-2 spike antibody titers after the COVID-19 vaccine and impaired fasting glucose (IFG) and diabetes. Methods: Participants were 953 health care workers aged 21–75 years who were tested for SARS-CoV-2 spike IgG antibodies and underwent a health checkup two months after their second dose of the BNT162b2 vaccine. IFG was defined as a fasting plasma glucose (FPG) level of 100–125 mg/dL, and diabetes was defined as an FPG level ≥ 126 mg/dL or being under medical care for diabetes. Multivariable linear regression was used to calculate the ratio of the mean. Result: Spike IgG antibody titers were lower in the presence of hyperglycemia; the ratios of the means (95% CI) were 1.00, 0.79 (0.60–1.04), and 0.60 (0.42–0.87) for individuals with normoglycemia, IFG, and diabetes, respectively (*p* trend < 0.001). Restricted cubic spline regression analysis showed that IgG spike antibody titers decreased linearly with increasing concentrations of FPG. Conclusion: Diabetes and, to a lesser extent, IFG may be associated with poor humoral immune response after BNT162b2 vaccination.

## 1. Introduction

Coronavirus disease 2019 (COVID-19) has affected over 416 million people and caused over 5.1 million deaths worldwide as of 14 February 2022 [1]. The severity of COVID-19 has been reported to be three times higher in people with diabetes [2]. Vaccines are considered the most important tool for curbing the rapid spread of COVID-19. As of 16 February 2022, 10.42 billion doses of COVID-19 vaccine have been administered globally across 218 countries [3]. In Japan, 79.3% of the population has received two doses of COVID-19 vaccine (Pfizer/BioNTech, Moderna, or AstraZeneca), as of 16 February 2022 [3].

Experimental data show that diabetes interferes with the activation of innate and acquired immunity [4,5]; thus, vaccine efficacy for those with diabetes is of great concern. Randomized controlled trials [6,7] have reported that COVID-19 vaccines are highly effective in preventing COVID-19, irrespective of diabetes status. In a recent study using real-world data, however, vaccine efficacy among people with diabetes was found to be somewhat lower than that among the total population [8]. In this context, studies of the effect of diabetes on immune response after vaccination would provide valuable data in conjunction with vaccine efficacy data.

A systematic review of eight (observational, case–control, and cross-sectional) studies reported a consistently lower antibody titer or seropositivity of COVID-19 vaccines that was associated with diabetes or poor diabetes control [9]. Subsequently, studies in Kuwait [10] and Japan [11] also reported lower concentrations of SARS-CoV-2 IgG spike antibody in patients with diabetes [10,11], whereas another Japanese study reported no significant association [12]. Most of these studies assessed diabetes based on participants’ self-report [9,10,12], which may miss undiagnosed diabetes. Along with diabetes, impaired fasting glucose (IFG) has been shown to be associated with immunologic abnormalities [13,14]. To the best of our knowledge, no studies have assessed the correlation between IFG and SARS-CoV-2 vaccine-induced antibody production.

To address these issues, we examined the associations of IFG and diabetes with SARS-CoV-2 spike antibody titers among Japanese health care workers who received two doses of the BNT162b2 vaccine. We hypothesized that participants with IFG and diabetes would have lower SARS-CoV-2 spike antibody titers than those of participants with normoglycemia.

## 2. Methods

### 2.1. Study Design

A repeat serological survey was launched among the staff of the National Center for Global Health and Medicine, Tokyo, Japan (NCGM) to monitor the spread of SARS-CoV-2 infection. The details of the study design are available elsewhere [15]. Participants were asked to donate venous blood and complete a questionnaire including queries regarding COVID-19 (vaccination history, history of COVID-19, etc.) and health-related lifestyles. Written informed consent was obtained from each participant, and the study procedure was approved by the NCGM ethics committee.

### 2.2. Participants

We used data from the third survey in June 2021, two months after the in-house vaccination program (COVID-19 mRNA-LNP BNT162b2; Pfizer-BioNTech). Of 3072 workers invited, 2779 (90%) participated. Of these, 2479 participants had received two doses of vaccine. Because data on the timing of blood draw (fasting or nonfasting) were not available for the participants in Kohnodai hospital ward (*n* = 528), we included participants in Toyama hospital ward (*n* = 1951). Of these, we excluded those who completed the survey within 14 days of the second vaccination (*n* = 2) and those with a history of COVID-19 (*n* = 10). We then excluded those with missing data on plasma glucose (*n* = 394) or plasma glucose measured in nonfasting conditions (*n* = 592), except for those with self-reported diabetes, leaving 953 participants (aged 21–75 years) for analysis.

### 2.3. Assessment of IFG and Diabetes

Plasma blood glucose was measured using an enzymatic (Hexokinase UV) method (Cica Liquid GLU, Kanto Chemical Co., Tokyo, Japan). Following the American Diabetes Association criteria [16], we defined IFG as a fasting plasma glucose (FPG) level of 100–125 mg/dL and diabetes as an FPG level of ≥126 mg/dL or being under medical care for diabetes. As a sensitivity analysis, we also defined IFG as an FPG of 110–125 mg/dL according to the World Health Organization (WHO) criteria [17]. Normoglycemia was defined as values below the cutoff point for IFG for each diagnostic criterion.

### 2.4. Measurements of SARS-CoV-2 Spike Antibody Titers

IgG against the SARS-CoV-2 spike protein was detected by performing an AdviseDxSARS-CoV-2 IgG II assay using Abbott ARCHITECT^®^ following the manufacturer’s instructions [18]. The assay detects the IgG antibodies against the receptor-binding domain (RBD) of the S1 subunit of the SARS-CoV-2 spike protein using chemiluminescent microparticle immunoassay (CMIA). The resulting chemiluminescence in relative light units (RLU) indicates the strength of the response, which in turn reflects the quantity of IgG-S present.

### 2.5. Assessment of Covariates

Age [11,19,20], sex [19], body mass index (BMI) [21], hypertension [11,20], smoking [19,20], alcohol consumption [22], and days after second vaccine [20] have been shown to be associated with SARS-CoV-2 vaccine antibody; we thus included those factors as covariates in analysis. BMI was computed as weight in kilograms divided by height in meters squared. Hypertension was defined as systolic blood pressure ≥ 140 mmHg, diastolic blood pressure ≥ 90 mmHg, or treatment for hypertension. Alcohol consumption (averaged daily consumption) was estimated by consumption frequency and amount consumed per day and expressed in go (180 mL) per day.

### 2.6. Statistical Analysis

Proportions and means were presented to show the background characteristics of the study population according to glycemic status. We transformed IgG spike antibody titers in a log scale before analysis. Multivariable linear regression analyses were used to calculate the mean and 95% confidence interval (CI) of the log-transformed IgG spike antibody titer across normoglycemia, IFG, and diabetes. In the first model, we adjusted for age (year, continuous) and sex. In the second model, we made additional adjustments for BMI (kg/m^2^, continuous), cigarette smoking (yes or no), alcohol drinking (nondrinker, occasional drinker, <1 go/day, or ≥1 go/day), hypertension (yes or no), and days after the second vaccination (days, continuous). The estimates obtained were then back-transformed to present the geometric mean titer (95% CI) and the ratio of the mean (95% CI). Trends were tested by assigning ordinal numbers to the categories of normoglycemia, IFG, and diabetes and treating them as a continuous variable. We also created restricted cubic spline plots to explore the shape of the association between FPG levels and IgG antibody titers, fitting a restricted cubic spline function with 3 knots placed at the 10th, 50th, and 90th percentiles [23]. Two-sided *p* values (<0.05) were regarded as statistically significant. All analyses were performed using the statistical software Stata version 17 (StataCorp, College Station, TX, USA).

## 3. Results

Table 1 shows the participants’ characteristics according to diabetic status (ADA criteria). The percentages of participants with IFG and diabetes were 5.0% and 2.2%, respectively. Compared with the participants with normoglycemia, participants with IFG and diabetes were older, were more likely to be male and current smokers, had higher means of BMI, a higher prevalence of hypertension, and had a longer interval between the second vaccination and blood sampling. Participants with IFG were more likely to be an alcohol drinker compared with the participants with normoglycemia and diabetes.

As shown in Table 2, participants with glucose metabolism abnormality had a lower IgG spike antibody titer than did the participants with normoglycemia. The multivariable-adjusted estimated geometric means (95% CI) were 5530 AU/mL (5301–5770), 3353 AU/mL (2348–4790), and 4374 AU/mL (3337–5733) for participants with normoglycemia, IFG, and diabetes, respectively. The multivariable-adjusted ratio of means (95% CI) were 1.00, 0.79 (0.60–1.04), and 0.60 (0.42–0.87) for participants with normoglycemia, IFG, and diabetes, respectively (*p* trend < 0.001). This association was materially unchanged when IFG was defined according to the WHO criteria (Appendix A). In cubic spline regression analysis (Figure 1), IgG spike antibody titers decreased linearly with increasing concentrations of FPG above 90 mg/dL.

## 4. Discussion

In the present study among health care workers who completed two doses of the BNT162b2 vaccine, participants with diabetes had a 40% lower mean of SARS-CoV-2 spike IgG antibodies than did the participants with normoglycemia. IFG was also associated with decreased titers of SARS-CoV-2 spike IgG antibody, although the association was not statistically significant. Restricted cubic spline regression analysis showed a decreasing trend of SARS-CoV-2 spike IgG antibody titers with increasing concentrations of FPG above 90 mg/dL.

The present findings for diabetes are consistent with the conclusions of the systematic review of eight studies with different designs. Specifically, four studies compared IgG spike antibody titer quantitively between individuals with diabetes and those without (similar to the present study), three studies compared IgG positivity between the two groups, and one study among patients with diabetes compared neutralizing antibody titer between those with poor glycemic control and those with good glycemic control [9]. We are aware of three additional studies that measured SARS-CoV-2 IgG spike antibody titers quantitively and were not included in that systematic review [10,11,12]; of these, two reported a significant reduction in SARS-CoV-2 IgG spike antibody titers in patients with diabetes [10,11]. In short, the current evidence linking diabetes to a lower immune response to the COVID-19 vaccine is quite consistent despite the differences in study design. Of the seven previous studies that quantified IgG spike antibody titer [10,11,12,19,20,24,25], six studies defined diabetes based on self-report only [10,12,19,20,24,25]. With the use of FPG to define diabetes and adjustment for several potential confounders, the present study provides more robust data to support the detrimental effect of diabetes on the production of antibodies after BNT162b2 vaccination.

Although the precise biological mechanism linking diabetes to vaccine-induced immunogenicity remains unclear, hyperglycemia induces various immune defects, including the dysfunction of monocyte/macrophage and neutrophil function, the suppression of cytokine productions, and the inhibition of complement activation, which in turn inhibits the production of antibodies after vaccination [4,5].

We found 21% lower mean IgG spike antibody titers among people with IFG relative to those with normoglycemia, although this was not statistically significant. To the best of our knowledge, the present study is the first to demonstrate a link between IFG and vaccine-induced antibody production. The present finding is supported by data from biomarker studies. Compared with individuals with normoglycemia, those with IFG had higher serum concentrations of proinflammatory cytokine interleukin 6 [13,14], which is a sign of an impaired immune system [26]. Additionally, we found a linear, inverse relationship between FPG levels and SARS-CoV-2 spike IgG antibody titers above an FPG of 90 mg/dL, suggesting that keeping lower concentrations of FPG within the nondiabetic range is preferable for enhancing vaccine-induced antibody production.

Some study limitations warrant mention. First, we did not assess cellular immune response, another key mechanism of infection protection [27]. Second, SARS-CoV-2 anti-spike IgG antibody does not fully characterize the humoral response induced by the vaccine. Nevertheless, spike antibody titers measured with the assay we employed were well correlated with neutralizing antibody titers in a subgroup of vaccine recipients in this cohort (spearmen’s ρ = 0.91) [18]. Third, an association observed in a cross-sectional study does not necessarily indicate causality. Fourth, although we adjusted for a wide range of potential factors, the possibility of residual confounding cannot be ruled out. Finally, the study participants were apparently healthy workers in a single medical facility. Caution should be exercised in generalizing the present findings to populations with different backgrounds.

In conclusion, vaccine recipients with diabetes and IFG had lower concentrations of SARS-CoV-2 spike IgG antibody than the vaccine recipients with normoglycemia did. The results of the present study inform researchers of the need for the careful monitoring of vaccine efficacy among individuals with hyperglycemia, including those in the nondiabetic range.

## Figures and Tables

**Figure 1 vaccines-10-00776-f001:**
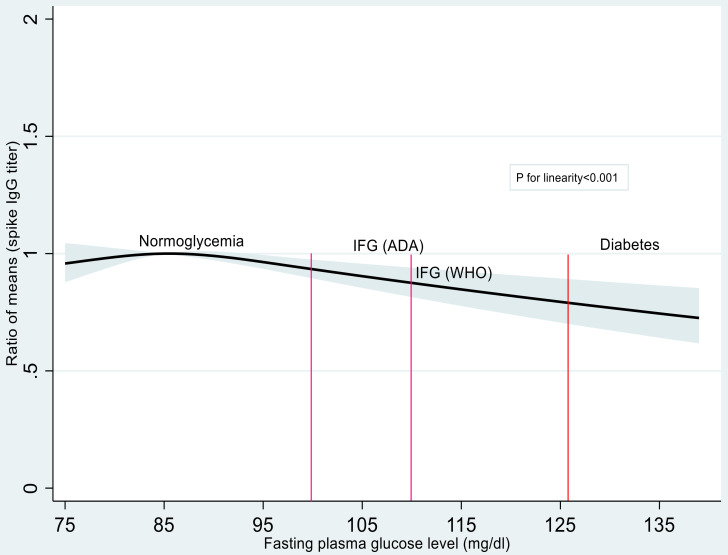
Restricted cubic spline regression for the association between the fasting plasma glucose and SARS-CoV-2 spike antibody titers. The solid line represents the ratios of means, and the bluish-gray area represents the 95% CI (linear trend, *p* < 0.001). Knots were placed at the 10th, 50th, and 90th percentiles (76, 85, and 96.9 mg/dL) of the fasting plasma glucose. The reference value was 85 mg/dL. The model was adjusted for age (year, continuous), sex, BMI (kg/m^2^, continuous), cigarette smoking (yes or no), alcohol drinking (nondrinker, occasional drinker, <1 go/day, or ≥1 go/day), hypertension (yes or no), and days after the second vaccination (days, continuous). IFG, impaired fasting glucose; ADA, American Diabetes Association; WHO, World Health Organization.

**Table 1 vaccines-10-00776-t001:** Participants’ characteristics across glycemic status.

Characteristics	Normoglycemia (N = 884)	IFG (N = 48)	Diabetes (N = 21)
Age (mean ± SD, year)	31.4 ± 12.0	44.2 ± 12.3	43.8 ± 12.4
Men (%)	29.2	62.5	52.4
Body mass index (mean ± SD, kg/m^2^)	21.4 ± 3.1	24.2 ± 3.9	25.5 ± 3.9
Smoker (%)	4.2	6.2	14.3
Alcohol drinker (≥1 go/day, %)	10.5	27.1	9.5
Hypertension (%)	6.4	37.5	52.4
Interval between the 2nd dose of vaccine and antibody test [median (IQR), days]	67 (61 to 70)	68.5 (62.5 to 71)	69 (66 to 71)

IFG, impaired fasting glucose; SD, standard deviation; IQR, interquartile range.

**Table 2 vaccines-10-00776-t002:** Multivariable-adjusted estimated geometric means (GMT) (95% CI) and ratio of mean (95% CI) of SARS-CoV-2 spike antibody titers according to the glycemic status *.

	SARS-CoV-2 Spike IgG Antibodies	
	GMT (95% CI)	Ratio of Mean (95% CI)
Normoglycemia	5530 (5301–5770)	1.00 (Reference)
IFG	4374 (3337–5733)	0.79 (0.60–1.04)
Diabetes	3353 (2348–4790)	**0.60 (0.42–0.87)**
*p*^§^ trend		**<0.001**

IFG, impaired fasting glucose; FPG, fasting plasma glucose; CI, confidence interval. Values in bold are statistically significant. Model was adjusted for age (years, continuous), sex, BMI (kg/m^2^, continuous), cigarette smoking (yes or no), alcohol drinking (nondrinker, occasional drinker, <1 go/day, or ≥1 go/day), hypertension (yes or no), and days after the second vaccination (days, continuous). * Normoglycemia was defined as <100 mg/dL, IFG was defined as FPG 100–125 mg/dL, and diabetes was defined as FPG ≥126 mg/dL or being under medical care for diabetes. ^§^ Based on linear regression analysis assigning ordinal numbers to the normoglycemia, IFG, and diabetes status.

## Data Availability

The data presented in this study are available on request from the corresponding author.

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
