# Peer review of "Association of Impaired Fasting Glucose and Diabetes with SARS-CoV-2 Spike Antibody Titers after the BNT162b2 Vaccine among Health Care Workers in a Tertiary Hospital in Japan"

_vaccines, 2022, doi:10.3390/vaccines10050776_

Round 1

Reviewer 1 Report

The present research measures anti-SARS CoV-2 spike antibodies in apparently healthy individuals, diabetic patients, and patients with impaired fasting glucose (IFG). The authors identified decreased antibodies in diabetic patients (statistically significant) and to a less extend to patients with IFG (not statistically significant).  The authors admit that there are seven previous studies that quantify IgG spike antibody. However, since six of them are based on self-report for diabetes incidence, their research (based on IFG measurement) provides more robust data to support the effect of Diabetes on the production of SARS-CoV-2 spike antibodies after vaccination. However, one of the previous studies (reference 11) not based on self-report, correlated antibodies concentration with Hb1c, measured by the researchers and came to analogue results with the present study.

All previews studies came to similar conclusions with the exception of one which reported no significant association. So, the contribution of the present study in new knowledge is pure. However, it is the only study which focused in the effect of glucose impairment, revealing a linear negative correlation between fasting plasma glucose and spike IgG ratio (compared to normoglycemia).

I propose acceptance after minor revision.

Observations:

In Introduction

Page 2, lane 11: The authors write: “To our knowledge, no studies assessed the impact of IFG on the production of SARS-CoV-2 vaccine-induced antibody” I would prefer the sentence to be replaced with “To our knowledge, no studies assessed the correlation between IFG and SARS-CoV-2 vaccine-induced antibody production”. The first sentence implies causality. But as authors admit in Discussion, an association observed in a cross-sectional study does not necessarily indicate causality.

In Methods

2.1. Study design, lane 3: The authors write “In brief, anti-SARS-CoV-2 nucleocapsid (all surveys) and spike protein antibodies (from the second survey onward) were measured.” However, no where in the study the measurement of nucleocapsid antibodies were mentioned or used. Is something missing or the sentence has to be omitted?

Author Response

We have studied all the comments carefully, described our responses (in blue) point by point to each comment (in black), and revised our manuscript according to the reviewers’ comments.

Reviewer 1

The present research measures anti-SARS CoV-2 spike antibodies in apparently healthy individuals, diabetic patients, and patients with impaired fasting glucose (IFG). The authors identified decreased antibodies in diabetic patients (statistically significant) and to a less extend to patients with IFG (not statistically significant).  The authors admit that there are seven previous studies that quantify IgG spike antibody. However, since six of them are based on self-report for diabetes incidence, their research (based on IFG measurement) provides more robust data to support the effect of Diabetes on the production of SARS-CoV-2 spike antibodies after vaccination. However, one of the previous studies (reference 11) not based on self-report, correlated antibodies concentration with Hb1c, measured by the researchers and came to analogue results with the present study.

All previews studies came to similar conclusions with the exception of one which reported no significant association. So, the contribution of the present study in new knowledge is pure. However, it is the only study which focused in the effect of glucose impairment, revealing a linear negative correlation between fasting plasma glucose and spike IgG ratio (compared to normoglycemia).

I propose acceptance after minor revision.

We greatly appreciate the reviewers’ thoughtful comments on the manuscript. We have carefully revised our manuscript according to the comments. Our responses to all the comments are as follows:

In Introduction

Page 2, lane 11: The authors write: “To our knowledge, no studies assessed the impact of IFG on the production of SARS-CoV-2 vaccine-induced antibody” I would prefer the sentence to be replaced with “To our knowledge, no studies assessed the correlation between IFG and SARS-CoV-2 vaccine-induced antibody production”. The first sentence implies causality. But as authors admit in Discussion, an association observed in a cross-sectional study does not necessarily indicate causality.

Response: Thank you for the valuable suggestions. Following your suggestion, we revised the sentence, as follows—

(Line 25-26) “To our knowledge, no studies assessed the correlation between IFG and SARS-CoV-2 vaccine-induced antibody production. …”

In Methods

2.1. Study design, lane 3: The authors write “In brief, anti-SARS-CoV-2 nucleocapsid (all surveys) and spike protein antibodies (from the second survey onward) were measured.” However, no where in the study the measurement of nucleocapsid antibodies were mentioned or used. Is something missing or the sentence has to be omitted?

Response: Although we measured the anti-SARS-CoV-2 nucleocapsid in our study, we did not use the information for the present study. So, we deleted the sentence in the revised version to avoid confusion.

Reviewer 2 Report

This is a well-designed study, however, focusing on just SARSCoV-2 spike antibody titers limits the study. I suggest running another assay such as antibodies to tetanus, diphtheria, or measles as a control for this study.
Minor changes :
  1. "Hyperglycemia can inhibit the activation of innate and acquired immunity" consider changing it to " Hyperglycemia can alter the activation of innate and acquired immunity".
  2. "To examine the association of impaired fasting glucose (IFG) and diabetes with SARS-CoV-2 spike antibody titers after COVID-19 vaccine." The question is about the antibody, so better use "To examine the association between SARS-CoV-2 spike antibody titers after COVID-19 vaccine and impaired fasting glucose (IFG) and diabetes".
  3. In methods:  - 2.2. Participants "For the present study," No need for this statement.
  4. Plasma blood glucose was measured using an enzymatic (Hexokinase UV) method. Please add the name of the kit and supplier information.

Author Response

Thank you very much for your very careful review. We have studied all the comments carefully, described our responses (in blue) point by point to each comment (in black), and revised our manuscript according to the reviewers’ comments.

Reviewer 2

We would like to thank the reviewer for carefully reviewing our manuscript. We have carefully revised our manuscript and hope that the revised version and the responses we provided below satisfactorily address all the issues that reviewer had noted.

This is a well-designed study, however, focusing on just SARSCoV-2 spike antibody titers limits the study. I suggest running another assay such as antibodies to tetanus, diphtheria, or measles as a control for this study.

Response: It would be of great interest to see if impaired glucose metabolism is associated with antibodies to diseases other than COVID-19. However, our study protocol (focusing on COVID-19) does not allow us to measure antibodies to tetanus, diphtheria, etc (even as a control). Please understand the scope of our study.
However, we should acknowledge the lack of humoral immune response as a study limitation. Therefore, we revised our manuscript by adding one limitation—

 “First, we did not assess cellular immune response, another key mechanism of infection protection [27]. Second, SARS-CoV-2 anti-spike IgG antibody does not fully characterize the humoral response induced by the vaccine. Nevertheless, spike antibody titers measured with the assay we employed were well correlated with neutralizing antibody titers in a subgroup of vaccine recipients in this cohort (spearmen’s ρ=0.91) [18].”

"Hyperglycemia can inhibit the activation of innate and acquired immunity" consider changing it to " Hyperglycemia can alter the activation of innate and acquired immunity".

Response: Following your suggestion, we revised the first sentence of the background part of the abstract, as follows—

(Abstract)

“Background: Hyperglycemia can alter the activation of innate and acquired immunity, but epidemiological evidence linking hyperglycemia to post-vaccination immunogenicity is limited.”

"To examine the association of impaired fasting glucose (IFG) and diabetes with SARS-CoV-2 spike antibody titers after COVID-19 vaccine." The question is about the antibody, so better use "To examine the association between SARS-CoV-2 spike antibody titers after COVID-19 vaccine and impaired fasting glucose (IFG) and diabetes".

Response: As reviewer suggested, we revised the sentence of the objective part of the abstract, as follows—

(Abstract)

“Objective: To examine the association between SARS-CoV-2 spike antibody titers after COVID-19 vaccine and impaired fasting glucose (IFG) and diabetes.”

In methods:  - 2.2. Participants "For the present study," No need for this statement.

Response: Following your suggestion, we deleted “for the present study” from the beginning of the participants part of the method section and revised it as follows—

 “We used data of the third survey in June 2021, two months after the in-house vaccination program (COVID-19 mRNA-LNP BNT162b2; Pfizer-BioNTech).”

Plasma blood glucose was measured using an enzymatic (Hexokinase UV) method. Please add the name of the kit and supplier information.

Response: As reviewer suggested, we added the name of the kit and supplier information, and revised the first sentence of the “assessment of IFG and diabetes” part—

“Plasma blood glucose was measured using an enzymatic (Hexokinase UV) method (Cica Liquid GLU, Kanto Chemical Co., Tokyo, Japan).”

Round 2

Reviewer 2 Report

The authors have addressed all previous comments.